



# Modulation of daily PM$_{2.5}$ concentrations over China in winter by large-scale circulation and climate change

Zixuan Jia[1], Carlos Ordóñez[2], Ruth M. Doherty[1], Oliver Wild[3], Steven T. Turnock[4,5], and Fiona M. O'Connor[4]

[1]School of GeoSciences, University of Edinburgh, Edinburgh, UK
[2]Departamento de Física de la Tierra y Astrofísica, Facultad de Ciencias Físicas, Universidad Complutense de Madrid, Madrid, Spain
[3]Lancaster Environment Centre, Lancaster University, Lancaster, UK
[4]Met Office Hadley Centre, Exeter, UK
[5]University of Leeds Met Office Strategic Research Group, School of Earth and Environment, University of Leeds, Leeds, UK

*Correspondence to*: Zixuan Jia (Z.Jia-6@sms.ed.ac.uk)

**Abstract.** We use the United Kingdom Earth System Model, UKESM1, to investigate the influence of the winter large-scale circulation on daily concentrations of PM$_{2.5}$ (particulate matter with an aerodynamic diameter of 2.5 µm or less) and their sensitivity to emissions over major populated regions of China over the period 1999–2019. We focus on the Yangtze River Delta (YRD), where weak flow of cold, dry air from the north and weak inflow of maritime air are particularly conducive to air pollution. These provide favourable conditions for the accumulation of local pollution but limit the transport of air pollutants into the region from the north. Based on the dominant large-scale circulation, we construct a new index using the north-south pressure gradient and apply it to characterize PM$_{2.5}$ concentrations over the region. We show that this index can effectively distinguish different levels of pollution over YRD and explain changes in PM$_{2.5}$ sensitivity to emissions from local and surrounding regions. We then project future changes in PM$_{2.5}$ concentrations using this index and find an increase in PM$_{2.5}$ concentrations over the region due to climate change that is likely to partially offset the effect of emission control measures in the near-term future. To benefit from future emission reductions, more stringent emission controls are required to offset the effects of climate change.

## 1 Introduction

Haze air pollution with high levels of PM$_{2.5}$ (particulate matter with an aerodynamic diameter of 2.5 µm or less) is a major health concern in China, especially in the major populated regions of Beijing–Tianjin–Hebei (BTH), the Yangtze River Delta (YRD) and the Pearl River Delta (PRD) (Zhao et al., 2013; Ding et al., 2013; Huang et al., 2014). Many studies have explored the underlying causes of high PM$_{2.5}$ concentrations, and air pollutant emissions (An et al., 2019; Chan and Yao, 2008; Zhang, et al., 2019a), meteorological conditions (Wang et al., 2009; Hou et al., 2018, 2020), and regional transport (Li et al., 2012; Sun et al., 2015; Chen et al., 2017; Wang et al., 2016) are identified as important contributors. In particular,



meteorological conditions can modulate the regional transport, as well as the local accumulation, chemical conversion and wet and dry deposition of air pollutants (e.g., Tai et al., 2010; Zhang et al., 2014; Wang et al., 2014).

Severe haze pollution frequently occurs in winter under stagnant meteorological conditions with weak near-surface winds,

strong temperature inversions and high relative humidity, which are favourable for the accumulation of $PM_{2.5}$ (Wang et al., 2014; Miao et al., 2015; Leung et al., 2018). These meteorological conditions are in turn affected by large-scale circulation patterns over China, dominated by the East Asian winter monsoon during winter with northerlies along the East Asian coast and southerlies from the South China Sea and the East China Sea (Chang et al., 2006; Wang and Chen, 2010; Wang and Lu, 2017). Global climate models can represent these large-scale circulation features better than regional meteorological

conditions that typically depend on subgrid scale processes (Chen et al., 2012; Zha et al., 2020; Xu et al., 2021). Because of this, many studies have investigated the modulation of large-scale winter circulation on $PM_{2.5}$ concentrations in China and proposed circulation-based indices. Among these studies, most of the focus has been placed on the BTH region which has the most severe $PM_{2.5}$ pollution or on parts of southern China (e.g., Wang et al., 2010; Jia et al., 2015; Jeong and Park, 2017; Zhang et al., 2019b). As the circulation patterns are more complex over eastern China, fewer studies have focused on this

region (e.g., Wang et al., 2016; Leung et al., 2018; Hou et al., 2019). Recently, Jia et al. (2022) diagnosed the dominant large-scale circulation patterns associated with winter $PM_{2.5}$ and defined new circulation-based indices for BTH, YRD and PRD. However, these results were based on a short five-year period and need to be verified over a longer time period.

The sensitivity of air pollution to emission sources is associated with regional transport, which can be modulated by the

large-scale circulation. $PM_{2.5}$ originates from local emissions of primary particles emitted directly from natural and anthropogenic sources and from secondary particles generated by heterogeneous and homogeneous chemical reactions of gaseous precursors in the atmosphere (Feng et al., 2012; He et al., 2012; Du et al., 2020). Air pollution from surrounding regions also affects $PM_{2.5}$ concentrations through regional transport (Sun et al., 2015; Wang et al., 2016; Cheng et al., 2019; Sun et al., 2022). Cheng et al. (2019) found that the reduction in $PM_{2.5}$ concentrations in Beijing from 2013 to 2017, which

resulted from the implementation of an action plan for controlling anthropogenic emissions, was dominated by emission reductions from both local (65%) and surrounding regions (23%). In a heavy wintertime air pollution episode over the two central Chinese provinces of Hubei and Hunan in January 2019, 71% of the $PM_{2.5}$ concentrations were attributed to regional transport from northern China (Hu et al., 2021). YRD is a key emission source and receptor region in eastern China that is affected by both northerly continental winds from Siberia and southerly oceanic winds in winter (e.g., Li et al., 2012; Wang

et al., 2016; Jeong and Park, 2017). Consequently, emissions from the major source regions located north and south of the YRD, i.e., BTH and PRD, have the potential to affect air pollution in the YRD region (e.g., Zhao et al., 2013; Zhang and Cao, 2015; Liao et al., 2015). A combination of $PM_{2.5}$ formation from local emissions and regional transport from surrounding regions results in complex air pollution characteristics in YRD. It is therefore important to investigate the role of





emissions from local and surrounding regions in PM$_{2.5}$ pollution in the region during winter and to identify the impact of large-scale circulation.

Future PM$_{2.5}$ concentrations will be influenced by changes in both air pollutant emissions and climate. Circulation-based indices derived from climate models are commonly used to represent the future evolution of PM$_{2.5}$ concentrations (e.g., Cai et al., 2017; Zhao et al., 2021). For instance, more frequent severe haze days have been projected in Beijing under climate change based on the East Asian winter monsoon index (Pei et al., 2018) and the Haze Weather index (Cai et al., 2017). However, most existing circulation-based indices have been proposed for the North China Plain and do not reflect the link between the large-scale circulation and PM$_{2.5}$ levels over YRD and PRD. Furthermore, analyses of output from climate models point to large uncertainties in the magnitude and spatial extent of projections of circulation features over China during winter under climate change (Ding et al., 2007; Xu et al., 2016; Miao et al., 2020). Therefore, improved knowledge of the dominant large-scale circulation patterns and identification of appropriate circulation-based indices are needed to understand PM$_{2.5}$ concentration changes in future climate projections.

In this study we use a state-of-the-art Earth system model (Sects. 2, 3) to investigate the dominant large-scale circulation– PM$_{2.5}$ relationships for BTH, YRD and PRD on daily timescales during winter, and propose a new circulation-based index for each region (Sect. 4). We then quantify the sensitivity of daily PM$_{2.5}$ in YRD to emissions from local and surrounding regions and explain the modulation of this sensitivity by the large-scale circulation using the proposed new daily circulation-based index (Sect. 5). Based on these modelled dominant large-scale circulation–PM$_{2.5}$ relationships and circulation-based indices derived from climate model historical and future simulations, we project daily changes in PM$_{2.5}$ concentrations under climate change (Sect. 6). Finally, Sect. 7 summarises the main results.

## 2 Data and methodology

### 2.1 Model description and simulations

The United Kingdom Earth System Model, UKESM1, as configured for the latest Coupled Model Intercomparison Project Phase model, CMIP6, (Sellar et al., 2019, 2020), is used to simulate the impact of large-scale circulation on daily PM$_{2.5}$ concentrations for CMIP6 historical (1999-2014) and future (2015-2019) periods. Here, UKESM1 is configured to simulate changes in the atmosphere and land only as used in the Atmospheric Model Intercomparison Project (AMIP; Eyring et al., 2016). The United Kingdom Chemistry and Aerosols model (UKCA; Morgenstern et al., 2009; O'Connor et al., 2014) is the atmospheric composition component of UKESM1, and includes the stratosphere–troposphere gas-phase chemistry scheme, StratTrop (Archibald et al., 2020) and the GLOMAP-mode aerosol scheme (Mann et al., 2010; Mulcahy et al., 2020). UKCA is coupled with the Global Atmosphere 7.1/Global Land 7.0 (GA7.1/GL7.0; Walters et al., 2019) configuration of the Hadley



Centre Global Environment Model version 3 (HadGEM3; Hewitt et al., 2011). The model version used here permits simulation of PM$_{2.5}$ concentrations at daily resolution.

We output PM$_{2.5}$ concentrations and meteorological fields from 1$^{st}$ Dec 1999 to 28$^{th}$ Feb 2019 at a horizontal resolution of
1.875° × 1.25° (approximately 140 km at mid latitudes). We then extract daily concentrations for 20 winters from 1$^{st}$ Dec 1999 – 28$^{th}$ Feb 2000 to 1$^{st}$ Dec 2018 – 28$^{th}$ Feb 2019 (hereafter referred to as DJF 1999–2018). In this study, DJF refers to December of the current year and January and February of the following year. Meteorological fields include zonal wind at 850 hPa (U850), meridional wind at 850 hPa (V850), sea level pressure (SLP), precipitation and geopotential height at 500 hPa (Z500). Wind speed and temperature are nudged with ERA-Interim reanalyses from the European Centre for Medium-
Range Weather Forecasts (ECMWF) every 6 h (Dee et al., 2011). By nudging to reanalysis data this model can produce a realistic representation of the meteorological conditions. Sea surface temperature and sea ice fields are prescribed with observations from the National Oceanic and Atmospheric Administration (NOAA) (Reynolds et al., 2007). Greenhouse gas concentrations and vegetation land cover fraction are prescribed as in CMIP6 historical (1999-2014) and SSP3-7.0 future (2015-2019) simulations conducted by UKESM1 (Meinshausen et al., 2017, 2020). There is little difference between the
shared socio-economic pathways (SSPs; O'Neill et al., 2014; van Vuuren et al., 2014) in the first few years of each scenario, so the choice of scenario should not impact the results for the time period considered here. In order to isolate the meteorological contribution to daily PM$_{2.5}$ variability from emissions, CMIP6 emissions for 2014 are used for the full period of the simulations, 1999-2019. This allows robust relationships between PM$_{2.5}$ and the dominant large-scale circulation for BTH, YRD and PRD to be established. Furthermore, three additional simulations have been performed for a 6-year period
(2014-2019) with reduced emission fluxes over specific regions to examine the sensitivity of PM$_{2.5}$ over YRD to different source regions (Table 1).

To project changes in circulation-based indices and in PM$_{2.5}$ concentrations under climate change, we use daily meteorological data for DJF 1995-2098 derived from the CMIP6 UKESM1 historical experiment and future scenario SSP3-
7.0. The CMIP6 SSP3-7.0 scenario has a large anthropogenic climate forcing signal (a radiative forcing of 7.0 W m$^{-2}$ at 2100) and encompasses weak action on reducing air pollutant emissions (Turnock et al., 2020).

### 2.2 Reanalysis PM$_{2.5}$ data

The six-year high-resolution Chinese air quality reanalysis dataset (CAQRA; Kong et al., 2021) is the latest air quality reanalysis for China, and includes surface fields of PM$_{2.5}$ at high spatial (15 km×15 km) and temporal (1 h) resolution for the
period 2013–2018. CAQRA has been validated with independent observational datasets to reproduce the magnitude and variability of PM$_{2.5}$ in China on a regional scale (Kong et al., 2021). Furthermore, it has been used to investigate the modulation of daily air quality in China during winter by regional meteorological conditions and the large-scale circulation (Jia et al., 2022). We use PM$_{2.5}$ hourly concentrations from this dataset to calculate daily average PM$_{2.5}$ for DJF 2013-2017.



For evaluating the UKESM1 simulated daily $PM_{2.5}$ concentrations, we regrid the CAQRA reanalysis to the coarser spatial resolution of UKESM1.

## 3 Model evaluation of daily $PM_{2.5}$ concentrations

We have run the UKESM1 model for 1999-2019 with 2014 emissions. The extracted daily $PM_{2.5}$ concentrations are evaluated against $PM_{2.5}$ data from the CAQRA reanalysis. A short 3-month period (January-February-December of 2014, JFD 2014) has been used for a first comparison because UKESM1 emissions are fixed at 2014 levels. The spatial pattern of winter mean $PM_{2.5}$ concentrations in UKESM1 is broadly similar ($r = 0.87$, $p < 0.01$, $slope = 0.89$) to that found in CAQRA during JFD 2014, albeit with lower concentrations over most regions (Fig. 1). The model version used here does not include ammonium nitrate or formation of anthropogenic secondary organic aerosol, which may partly explain the underestimation of the $PM_{2.5}$ concentrations (Butt et al., 2017; Archibald et al., 2020). However, the results of this study should not be heavily impacted by this as we investigate the day-to-day variability of $PM_{2.5}$ concentrations rather than $PM_{2.5}$ concentrations directly. We define meteorologically coherent regions representing BTH, YRD and PRD by identifying UKESM1 grid cells where the simulated daily $PM_{2.5}$ concentrations are highly correlated ($r \geq 0.7$) with those representing Beijing (four grid cells), Shanghai (two grid cells) and Guangzhou (one grid cell), respectively (Fig. 2), following the approach of Jia et al. (2022). Daily regional $PM_{2.5}$ concentrations are then calculated by averaging grid cell concentrations over these three highly correlated homogeneous regions.

Figure 3a shows the evolution of the daily mean $PM_{2.5}$ concentrations from UKESM1 and CAQRA for DJF 2013-2017 over BTH, YRD and PRD, while Figure 3b compares their frequency distributions. The daily $PM_{2.5}$ concentrations from UKESM1 are significantly correlated ($p < 0.01$) with those from CAQRA ($r = 0.54$ over BTH; $r = 0.60$ over YRD; $r = 0.41$ over PRD). UKESM1 slightly overestimates the $PM_{2.5}$ concentrations in BTH likely because of the rapid emission reductions over the North China Plain after 2014. The underestimation found for the other two regions, especially in PRD, is consistent with the spatial distributions seen in Fig. 1, and cannot be attributed to emission controls. Overall, the spatial pattern of winter mean $PM_{2.5}$ concentrations (Fig. 1) and temporal evolution of daily $PM_{2.5}$ concentrations (Fig. 3) can be simulated well over all three regions, especially over YRD.

## 4 The impact of large-scale circulation on daily $PM_{2.5}$ concentrations

The wintertime large-scale circulation over East Asia is reproduced by UKESM1 by nudging with the ERA-Interim reanalysis. This is dominated by the Siberian High, as seen from the high SLP values centred over northwestern Mongolia, and by the Aleutian Low to its east and a low pressure over the Maritime Continent (hereafter referred to as the Maritime Continent Low) to its south (Fig. 4a). Cold and dry northwesterly lower tropospheric winds over northern China (Fig. 4a) are indicated by negative V850 values (Fig. 4b) and positive U850 values (4c). The middle tropospheric East Asian trough is characterised by low Z500 values over Northeast China as seen in Figure S1. Warm and wet southeasterly winds from the



South China Sea and the East China Sea are indicated by positive V850 values and negative U850 values, bringing precipitation over southern China (Fig. 4b-d). In this study, we investigate the influence of the large-scale circulation on the variability of daily $PM_{2.5}$ using 20 winters (DJF 1999-2018) from simulations.

We first examine the daily correlations of the $PM_{2.5}$ concentrations in each region with circulation variables and precipitation. The YRD region is shown as an example in Fig. 5 because the circulation patterns are more complex here (Fig. 4) and this region is less well studied than BTH and PRD as noted earlier. The patterns reveal that daily $PM_{2.5}$ concentrations in the YRD region are negatively correlated with SLP over northern China and positively correlated over southernmost China and the South China Sea (Fig. 5a). With regards to the wind components, correlations are positive with V850 over

northern China and negative over southern China and the South China Sea (Fig. 5b) as well as positive with U850 over eastern and central China (Fig. 5c). Furthermore, $PM_{2.5}$ concentrations in YRD are negatively correlated with precipitation over central and southeastern China (Fig. 5d). The comparison between the sign of these correlation coefficients and the winter mean patterns (Fig. 4) highlights the large-scale circulation features that are associated with high $PM_{2.5}$ pollution days over YRD. These days are mainly characterised by a weak Siberian High, a weak Maritime Continent Low, weak northerly

winds over northern China, weak southerly/easterly winds over southern/central China, and above-average westerly winds in eastern China. A weak Siberian High and weak Maritime Continent Low are identified as the dominant large-scale circulation features, because the largest coherent negative and positive correlation values are found for SLP over northern China and the South China Sea. The area-weighted averages of daily SLP over these two regions (yellow rectangles in Figure 5a) are significantly correlated ($p < 0.01$) with $PM_{2.5}$ concentrations in YRD of $r$ = -0.33 and $r$ = 0.32 respectively.

To reflect the effect of circulation over the Asian continent and the adjacent ocean, we use the SLP difference averaged over northern and southern China (i.e. [43–54°N, 102–122°E] minus [12–22°N, 95–111°E]) to build a north-south SLP gradient-based index for YRD ($I_{SLP\_YRD}$) for all days in DJF 1999-2018 (eq. 1). The daily SLP data are normalised by subtracting the mean and dividing by the standard deviation to yield a zero mean and unit variance before calculating the SLP gradient-

based index. Negative values of $I_{SLP\_YRD}$ indicate a weak pressure gradient between the Siberian High and Maritime Continent Low.

$$I_{SLP\_YRD} = \overline{SLP\ (43°-54°N, 102°-122°E)} - \overline{SLP\ (12°-22°N, 95°-111°E)} \qquad (1)$$

$I_{SLP\_YRD}$ is significantly correlated ($p < 0.01$) with $PM_{2.5}$ concentrations in YRD on daily time scales ($r$ = -0.47). These results point to a weak pressure gradient between the Siberian High and Maritime Continent Low as the dominant large-scale circulation pattern contributing to high $PM_{2.5}$ pollution in YRD. Comparing with the winter mean patterns (Fig. 4), on days with $I_{SLP\_YRD} < -1$ (Figs. 6a-d), a weak north-south pressure gradient inhibits the inflow of cold, dry northerly air to eastern China, creating appropriate conditions for the accumulation of aerosols and suppressing the southward transport of aerosols





away from YRD. Furthermore, a weak north-south pressure gradient also suppresses the inflow of warm, wet oceanic air from the East China Sea and the South China Sea. The associated reduction in precipitation is also likely to support high air pollution over YRD due to reduced wet deposition of $PM_{2.5}$ (e.g., Tai et al., 2010; Zhu et al., 2012; Leung et al., 2018). The reverse situation occurs on days with $I_{SLP\_YRD} > 1$ (Figs. 6e-h). Regional transport and wet deposition by precipitation have also been identified to contribute to the interannual variability in $PM_{2.5}$ concentrations over Shanghai in previous studies (e.g., Wang et al., 2016).

We have confirmed the capability of $I_{SLP\_YRD}$ to capture the relationship between the dominant large-scale circulation and daily $PM_{2.5}$ concentrations in YRD. To further examine the performance of $I_{SLP\_YRD}$ in distinguishing different levels of air pollution in the region, we compare the distributions of $I_{SLP\_YRD}$ for different percentile thresholds of daily $PM_{2.5}$ (Fig. 7). We group all winter days over the 20-year period below the 10th percentile (p10) of $PM_{2.5}$ concentrations as clean days (180 days), above the 90th percentile (p90) of $PM_{2.5}$ concentrations as heavily polluted days (180 days), between p10 and p50 (p10-50) as moderately clean days (720 days) and between p50 and p90 (p50-90) as moderately polluted days (720 days). The average value of $I_{SLP\_YRD}$ with associated 95% confidence intervals are: $I_{SLP\_YRD} = -0.53 \pm 0.10$ for heavily polluted days, $I_{SLP\_YRD} = -0.34 \pm 0.06$ for moderately polluted days, $I_{SLP\_YRD} = 0.20 \pm 0.05$ for moderately clean days and $I_{SLP\_YRD} = 1.09 \pm 0.10$ for clean days. These confidence intervals do not overlap, indicating that $I_{SLP\_YRD}$ can distinguish effectively between different levels of air pollution, and not just between heavily polluted and clean conditions. This SLP-gradient index, $I_{SLP\_YRD}$, improves on the capability of the SLP-based index derived by Jia et al. (2022) that considered only 5 years of winter data to distinguish $PM_{2.5}$ pollution levels in YRD. These results show that improved relationships between air pollution and the atmospheric circulation can be derived through the use of long-term modelled time series.

We conduct a similar analysis for the BTH and PRD regions (Figs. S2-S3 in the Supplement). A V850-based index over the East Asian coast (yellow rectangle in Figure S2c, [31–50°N, 113–124°E]) ($I_{V850\_BTH}$) and an SLP-based index over mainland China (yellow rectangle in Figure S3b, [23–42°N, 102–122°E]) ($I_{SLP\_PRD}$) are proposed for BTH and PRD, respectively, based on the largest coherent correlation values with $PM_{2.5}$ concentrations. As before, the meteorological fields have been averaged over the regions covered by those rectangles and normalised before the calculation of the indices. Both $I_{V850\_BTH}$ and $I_{SLP\_PRD}$ are significantly correlated ($p < 0.01$) with daily $PM_{2.5}$ concentrations in BTH ($r = -0.66$) and in PRD ($r = -0.57$) and can be used to distinguish $PM_{2.5}$ pollution levels. Our analyses suggest that $PM_{2.5}$ pollution in both regions is enhanced under suppressed northerly cold, dry winds over the East Asian coast (negative $I_{V850\_BTH}$) associated with a weakened Siberian High (negative $I_{SLP\_PRD}$) and a shallow East Asian trough at 500 hPa. Combined with the above analysis of the YRD region, weak transport of northerly cold, dry air as a consequence of a weakened Siberian High is identified to play an important role in air pollution accumulation in all three regions (i.e. YRD, BTH and PRD).



## 5 Influence of large-scale circulation on daily PM₂.₅ sensitivity to emissions

We have found that high PM$_{2.5}$ concentrations in YRD are associated with suppressed cold, dry air flow from the north and with reduced inflow of maritime air masses. These circulation patterns are associated with weak pressure gradients between the Siberian High and the Maritime Continent Low (negative values of $I_{SLP\_YRD}$). Such reduced pressure gradients may also

alter the transport of pollutants into the YRD from polluted regions to the north and south, and suppress the outflow of local pollutants from the region. To investigate this, we examine the sensitivity of PM$_{2.5}$ in YRD to local emissions and to those from surrounding upwind and downwind regions. We compare the results from 6 years of the nudged simulation (2014-2019 meteorology with emissions for year 2014) with those of three 6-year sensitivity simulations for the same period with reduced emissions over three regions: YRD (red grid cells in Figure 8), north China (top blue grid cells in Figure 8) and

south China (bottom blue grid cells in Figure 8) as shown in Table 1. In each of the sensitivity simulations, the main anthropogenic sources of PM$_{2.5}$ (sulphur dioxide, black carbon and organic carbon) are reduced to the values projected for the SSP3-7.0 pathway for 2058 over one of the three regions, to avoid using idealised changes. Accordingly, the winter mean emission flux of these sources of PM$_{2.5}$ decreases by 41% from 2014 (CMIP6 historical) to 2058 (CMIP6 SSP3-7.0) for all three regions (Table S1).

The winter daily mean PM$_{2.5}$ concentration in YRD during DJF 2014-2018 is 46.9 μg/m³ (average value over the red grid cells in Figure 8). After reducing emissions over YRD, north China and south China, the daily mean PM$_{2.5}$ concentration in YRD decreases by 3.1 μg/m³, 1.4 μg/m³ and 0.0 μg/m³, respectively. This indicates that PM$_{2.5}$ pollution over YRD mainly originates locally (69%) and, to a lesser extent, from north China (31%), as found in other studies (e.g., Li et al., 2012). We

then focus on heavily polluted days (PM$_{2.5}$ above p90) in YRD, for which the daily mean PM$_{2.5}$ concentration is 53.9 μg/m³ higher than the winter mean value. Because of this, at the same level of emission reduction, heavily polluted days are impacted more strongly, and daily mean PM$_{2.5}$ concentration in YRD decreases by 8.5 μg/m³, 3.4 μg/m³ and 1.0 μg/m³.

To examine the impact of the large-scale circulation on the contribution of emissions from local and surrounding regions, we

show the changes in PM$_{2.5}$ concentrations separately for all days during DJF 2014-2018 with $I_{SLP\_YRD} < -1$ and $I_{SLP\_YRD} > 1$ in Figure 9. On days with $I_{SLP\_YRD} < -1$, PM$_{2.5}$ pollution mainly accumulates over northern, central and eastern China, with PM$_{2.5}$ concentrations over YRD exceeding the winter mean by 14.3 μg/m³ (Fig. 9a). After reducing emissions over YRD, north China and south China, the daily mean PM$_{2.5}$ concentration in YRD decreases by 3.6 μg/m³, 1.4 μg/m³ and 0.2 μg/m³, respectively (Fig. 9b-d). The relative shares of the total reduction (5.2 μg/m³) from local (69%), north (27%) and south (4%) are similar to those for the winter mean. On days with $I_{SLP\_YRD} > 1$, the PM$_{2.5}$ concentrations over YRD are reduced to 16.6

μg/m³ below the winter mean (Fig. 9e). The reductions in mean PM$_{2.5}$ concentrations over YRD from local, north and south are 2.1 μg/m³ (57%), 1.5 μg/m³ (40%) and 0.1 μg/m³ (3%). The differences in the relative shares for days with $I_{SLP\_YRD} < -1$ and $I_{SLP\_YRD} > 1$ reflect the impact of the atmospheric circulation patterns on the sensitivity of PM$_{2.5}$ in YRD to emissions



from local and surrounding regions. Local emissions contribute more to PM$_{2.5}$ pollution over YRD for days with $I_{SLP\_YRD} <$ -1, because these days are associated with suppressed transport of pollutants and lower precipitation. Emissions from north China contribute more to PM$_{2.5}$ pollution over YRD for days with $I_{SLP\_YRD} > 1$, due to the southward aerosol transport to YRD on these days (e.g., Wang et al., 2016; Jeong and Park, 2017).

**6 Changes in circulation-based indices and PM$_{2.5}$ concentrations under climate change**

We have shown that our SLP gradient-based index ($I_{SLP\_YRD}$) performs well in capturing the dominant relationship between the large-scale circulation and PM$_{2.5}$ and in distinguishing PM$_{2.5}$ pollution levels in YRD. This suggests that $I_{SLP\_YRD}$ can serve as a robust indicator of PM$_{2.5}$ pollution over YRD, assuming that its variation is solely due to meteorology. In this section, we project future PM$_{2.5}$ concentrations over YRD, using $I_{SLP\_YRD}$ derived from UKESM1 CMIP6 data from the present day to
the end of the 21$^{st}$ century and the relationship between PM$_{2.5}$ concentrations and $I_{SLP\_YRD}$ derived from the nudged UKESM1 run for DJF 1999-2018.

We compare the winter daily values of $I_{SLP\_YRD}$ for the CMIP6 historical (1999-2014) period from nudged UKESM1 simulations with those calculated using data from the CMIP6 UKESM1 historical simulation. The frequency distributions of
daily mean $I_{SLP\_YRD}$ from these two datasets match well (Fig. S4 in the Supplement), suggesting that the CMIP6 UKESM1 simulations can be used to project $I_{SLP\_YRD}$ in the future under the SSP3-7.0 scenario. Compared to the present-day (1995-2014) mean, the projected $I_{SLP\_YRD}$ decreases gradually to mid-century (2039-2058) and to the end of century (2079-2098) (Fig. 10). This suggests a weakening of the pressure gradient between the Siberian High and the Maritime Continent Low under climate change. Furthermore, the interannual variability of $I_{SLP\_YRD}$, and therefore of the circulation patterns affecting PM$_{2.5}$, increase from present day until the end of the century. We have also examined changes in the circulation-
based indices identified for BTH and PRD (Fig. S5 in the Supplement). In contrast to $I_{SLP\_YRD}$, both $I_{V850\_BTH}$ and $I_{SLP\_PRD}$ exhibit little change from present day to the end of the century.

The simulated future decrease in $I_{SLP\_YRD}$ and the negative relationship between $I_{SLP\_YRD}$ and PM$_{2.5}$ concentrations over YRD
during DJF 1999-2018 found in Section 4 suggest that PM$_{2.5}$ concentrations over YRD are likely to increase in the future as a result of climate-driven changes in circulation in the absence of future emission changes. To verify this conjecture, we calculate the expected PM$_{2.5}$ concentrations for the different values of $I_{SLP\_YRD}$ in the nudged simulation for DJF 1999-2018 (Fig. S6 in the Supplement) and derive a relationship that can be used to estimate PM$_{2.5}$ in the CMIP6 UKESM1 simulations. As there is non-linearity and considerable spread, a resampling method is used. We first divide $I_{SLP\_YRD}$ into bins of 0.1 width
and then draw 10,000 values randomly from the set of PM$_{2.5}$ concentrations within each bin. The estimated PM$_{2.5}$ concentration in a given bin is calculated as the mean of the corresponding sample of 10,000 values. Winter daily PM$_{2.5}$ concentrations over YRD are calculated first by applying this relationship to $I_{SLP\_YRD}$ values derived from the CMIP6





UKESM1 historical simulation for 1999-2014. The mean value of these calculated winter daily PM$_{2.5}$ concentrations (48.4 µg/m$^3$) closely matches that directly diagnosed by nudged UKESM1 (48.3 µg/m$^3$).

We then use the same relationship to project winter daily PM$_{2.5}$ concentrations over YRD for present day (1995-2014), mid-century (2039-2058) and the end of century (2079-2098). The mean value of PM$_{2.5}$ concentration increases from present day (46.3 µg/m$^3$) to mid-century (48.9 µg/m$^3$) and to the end of the century (50.1 µg/m$^3$). This suggests that winter PM$_{2.5}$ concentrations over YRD will continue to increase until the end of century under the SSP3-7.0 scenario if there is no change in PM$_{2.5}$ emission sources. Although the SSP3-7.0 pathway encompasses some emission control measures on the main anthropogenic sources of PM$_{2.5}$ (Table S1), the expected air quality improvements are likely to be partially offset by an increase in PM$_{2.5}$ concentrations associated with a weaker pressure gradient between the Siberian High and the Maritime Continent Low. UKESM1 simulations from CMIP6 following the SSP3-7.0 pathway suggest that PM$_{2.5}$ concentrations in YRD are not expected to change substantially by mid-century. This indicates that the effects on PM$_{2.5}$ of the circulation changes we calculate due to climate change may still play an important role in the near term despite local emission changes. Nevertheless, by the end of the 21$^{st}$ century the benefits from emission reduction measures outweigh any penalties from circulation changes.

## 7 Conclusions

This study investigates the influence of the large-scale circulation on daily PM$_{2.5}$ concentrations and their sensitivity to local and regional emissions in China during winter. Using simulations with a state-of-the-art Earth system model (UKESM1) for DJF 1999-2018 with fixed 2014 emissions, we identify the dominant large-scale circulation patterns that display the strongest relationships with daily PM$_{2.5}$ concentrations in major populated regions of China (BTH, YRD and PRD), with a focus on YRD. The pressure gradient between the Siberian High and the Maritime Continent Low is found to have a strong relationship with daily PM$_{2.5}$ concentrations over YRD ($r = -0.47$). This negative correlation indicates that suppressed cold, dry air flow from the north and reduced inflow of maritime air associated with a weak north-south pressure gradient contribute to air pollution accumulation in the region. We therefore propose a new north-south sea level pressure gradient-based index for YRD ($I_{SLP\_YRD}$). There are few existing daily circulation indices defined for the region, and we demonstrate that $I_{SLP\_YRD}$ can explain the day-to-day variability of PM$_{2.5}$ concentrations and predict the occurrence of heavily polluted (PM$_{2.5}$ above p90), moderately polluted (PM$_{2.5}$ within p50–90), moderately clean (PM$_{2.5}$ within p10–50) and clean (PM$_{2.5}$ below p10) days.

By performing sensitivity simulations for DJF 2014-2018 with reduced emissions following the SSP3-7.0 scenario for 2058 over different regions, we find that local emissions contribute most to PM$_{2.5}$ pollution over YRD and that the sensitivity to emissions can be affected by the dominant large-scale circulation patterns. On days with $I_{SLP\_YRD} < -1$, a





weak pressure gradient, through reduced transport and precipitation, supports the accumulation of $PM_{2.5}$ from local emissions over the region. On days with $I_{SLP\_YRD} > 1$, a strong pressure gradient permits effective transport of northerly cold, dry air contributing to the inflow of air pollutants from north China.

5    Based on the simulated relationship between $PM_{2.5}$ and $I_{SLP\_YRD}$ with fixed emissions and the daily values of $I_{SLP\_YRD}$ derived from CMIP6 UKESM1 simulations, we project future changes in $PM_{2.5}$ concentrations over YRD. We find a decrease in $I_{SLP\_YRD}$ into the future. This suggests that winter mean climate-driven $PM_{2.5}$ concentrations over the region will increase over the century under the SSP3-7.0 pathway. We note, however, that emissions under SSP3-7.0 decrease over the YRD region, which should lead to reduced $PM_{2.5}$ levels. Overall, the calculated climate and emission driven

10   $PM_{2.5}$ concentrations from CMIP6 UKESM1 simulations change little by mid-century. Therefore, future changes in the large-scale circulation (i.e., a weaker pressure gradient between the Siberian High and the Maritime Continent Low) are likely to remain important in the near-term, partly offsetting any reduction in emissions. More stringent emission controls to offset climate change are required to ensure future $PM_{2.5}$ reductions along this pathway.

**Acknowledgements**

15   Oliver Wild and Ruth M. Doherty thank the Natural Environment Research Council (NERC) for funding under grants nos. NE/N006925/1, NE/N006976/1, and NE/N006941/1. Steven T. Turnock thanks the UK-China Research & Innovation Partnership Fund through the Met Office Climate Science for Service Partnership (CSSP) China as part of the Newton Fund. This work made use of computation resources on the Met Office and NERC joint supercomputer system (MONSooN) in the UK.



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





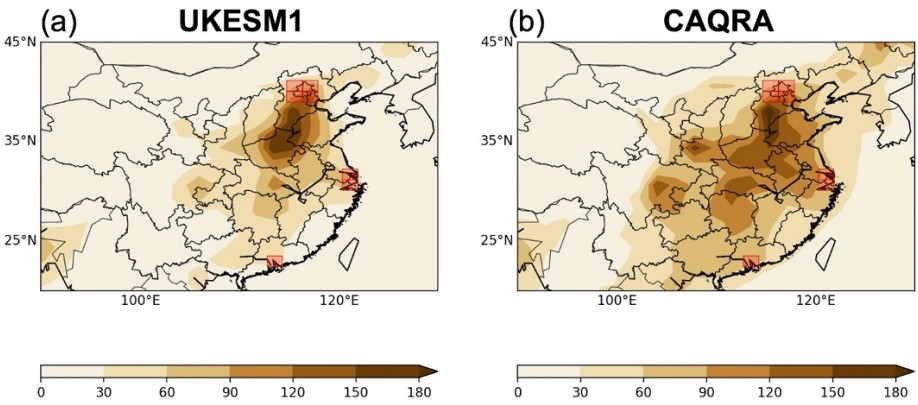

**Figure 1: Spatial distributions of winter mean daily PM$_{2.5}$ concentrations (μg/m$^3$) across China during JFD 2014 (a) simulated by UKESM1 and in (b) the CAQRA reanalysis. UKESM1 grid cells covering Beijing (four grid cells), Shanghai (two grid cells) and Guangzhou (one grid cell) are marked by red rectangles.**





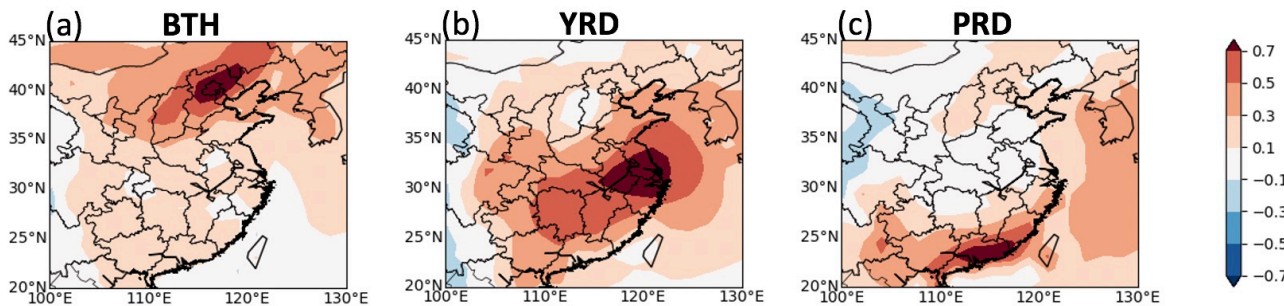

**Figure 2: Correlation coefficients of daily mean PM$_{2.5}$ concentrations over all UKESM1 grid cells with those in the grid cells covering (a) Beijing, (b) Shanghai and (c) Guangzhou during DJF 1999–2018. Regions where correlations are higher than 0.7 (dark red shading) are selected to represent the Beijing–Tianjin–Hebei (BTH), Yangtze River Delta (YRD) and Pearl River Delta (PRD) regions.**



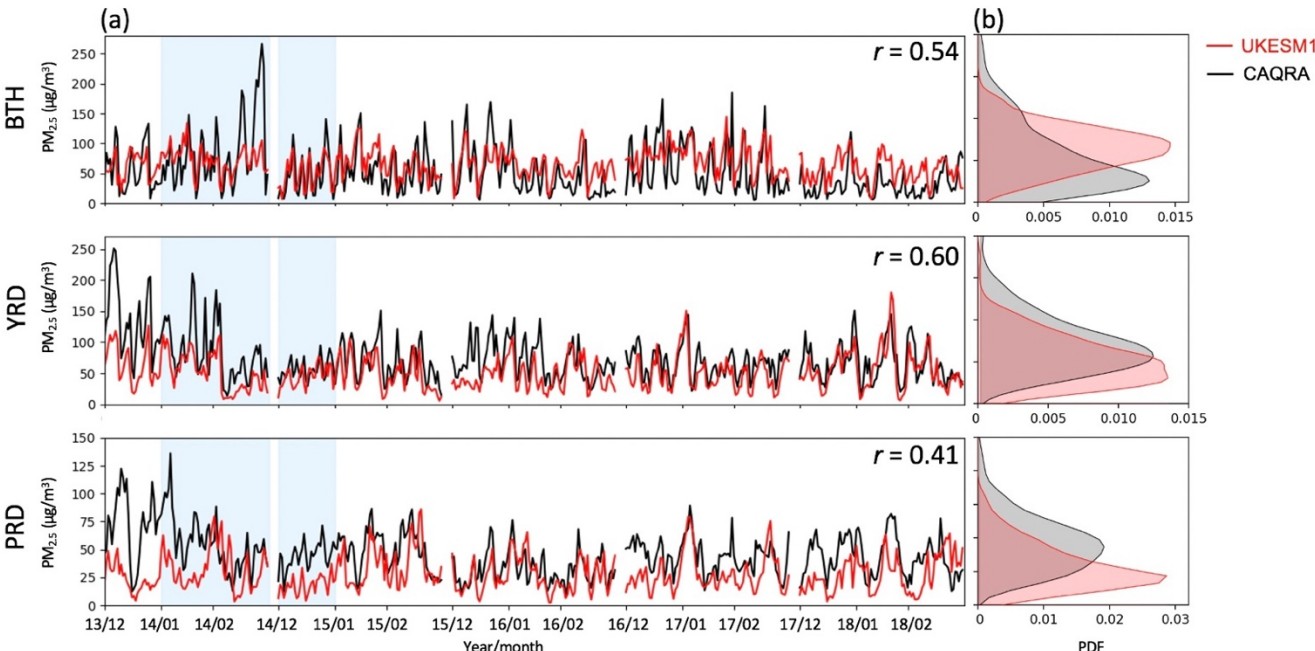

**Figure 3: Comparison of daily mean PM$_{2.5}$ concentrations (μg/m$^3$) provided by UKESM1 and CAQRA for BTH, YRD and PRD during DJF 2013–2017. (a) Time series of daily mean PM$_{2.5}$ concentrations. Blue area represents JFD 2014. The values of the Pearson's correlation coefficients ($r$) of the daily time series for all days in DJF 2013-2017 are also displayed. All correlation values are significant at the 99% confidence level using a two-tailed Student's $t$ test as indicated in von Storch and Zwiers (1999). (b) Frequency distributions of the daily mean PM$_{2.5}$ concentrations.**





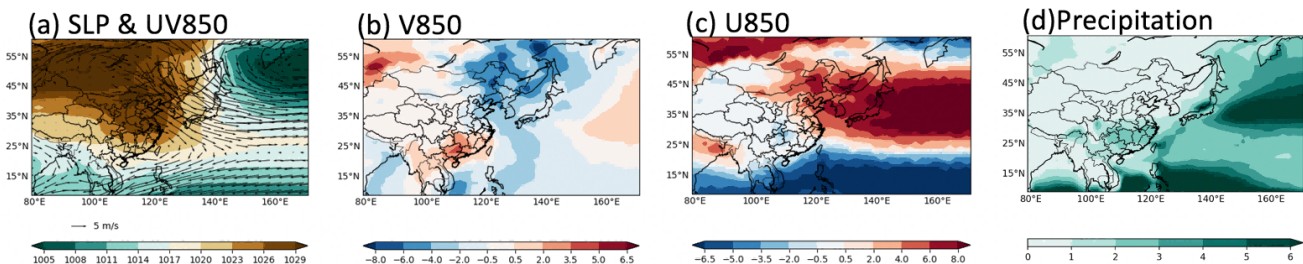

**Figure 4: Simulated winter mean daily (a) sea level pressure (SLP; hPa, shading) and 850 hPa wind (arrows), (b) 850 hPa meridional wind (V850; m/s), (c) 850 hPa zonal wind (U850; m/s), and (d) precipitation (mm/day) from nudged UKESM1 during DJF 1999–2018.**





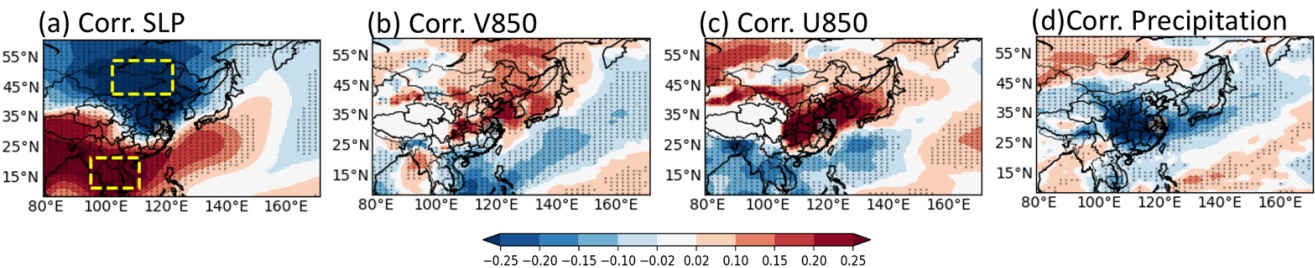

**Figure 5: Correlation coefficients of daily PM$_{2.5}$ concentrations in YRD with (a) SLP, (b) V850, (c) U850 and (d) precipitation from nudged UKESM1 during DJF 1999–2018. Dotted regions indicate significant correlations at the 95% level from a two-tailed Student's $t$-test. Grey shading represents the YRD region. The regions used for the definition of a circulation-based index for YRD (eq. 1) are marked by two yellow rectangles in panel a.**





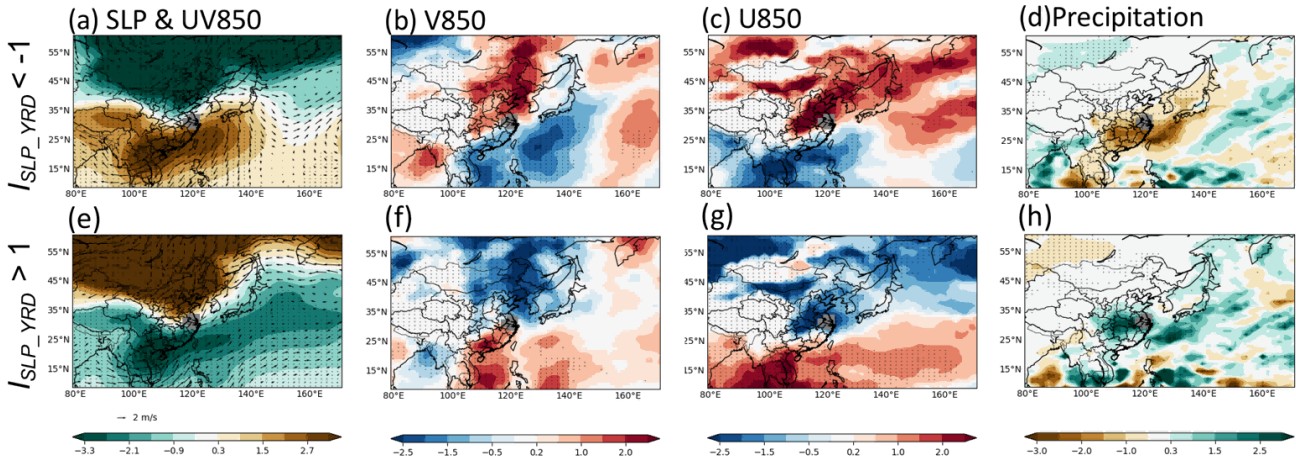

**Figure 6: Anomalies (days with $I_{SLP\_YRD}$ < -1 minus winter mean) of (a) sea level pressure (SLP; hPa, shading) and 850 hPa wind (arrows), (b) 850 hPa meridional wind (V850; m/s), (c) 850 hPa zonal wind (U850; m/s), and (d) precipitation (mm/day) from nudged UKESM1 during DJF 2014–2018, and anomalies (days with $I_{SLP\_YRD}$ > 1 minus winter mean) of (e) SLP, (f) V850, (g) U850 and (h) precipitation. Dotted regions mark statistically significant differences at the 95% level (determined through a bootstrap resampling method). Grey shading represents the YRD region.**





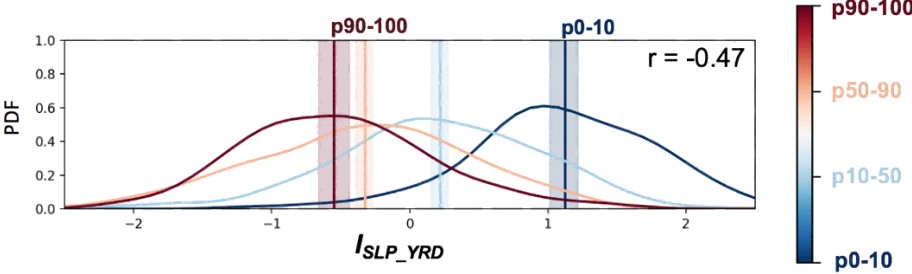

**Figure 7: Frequency distributions of a circulation-based index for YRD (eq. 1) for different percentile thresholds of daily mean PM$_{2.5}$ concentrations over YRD. Derived from nudged UKESM1 simulations during DJF 1999–2018. The vertical lines and shading represent the averages and the associated 95% confidence intervals, respectively. Averages are calculated using Tukey's trimean (e.g., Ge et al., 2019). The confidence intervals for these averages are estimated by using bootstrap resampling (e.g., Wang, 2001).**





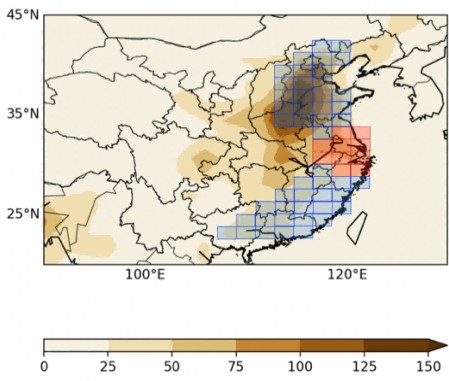

**Figure 8: Spatial distribution of the winter mean daily PM$_{2.5}$ concentrations (µg/m$^3$) simulated by UKESM1 across China during DJF 2014-2018 (shaded colours) and UKESM1 model grid cells representing north China (top blue grid cells), YRD (red grid cells) and south China (bottom blue grid cells).**



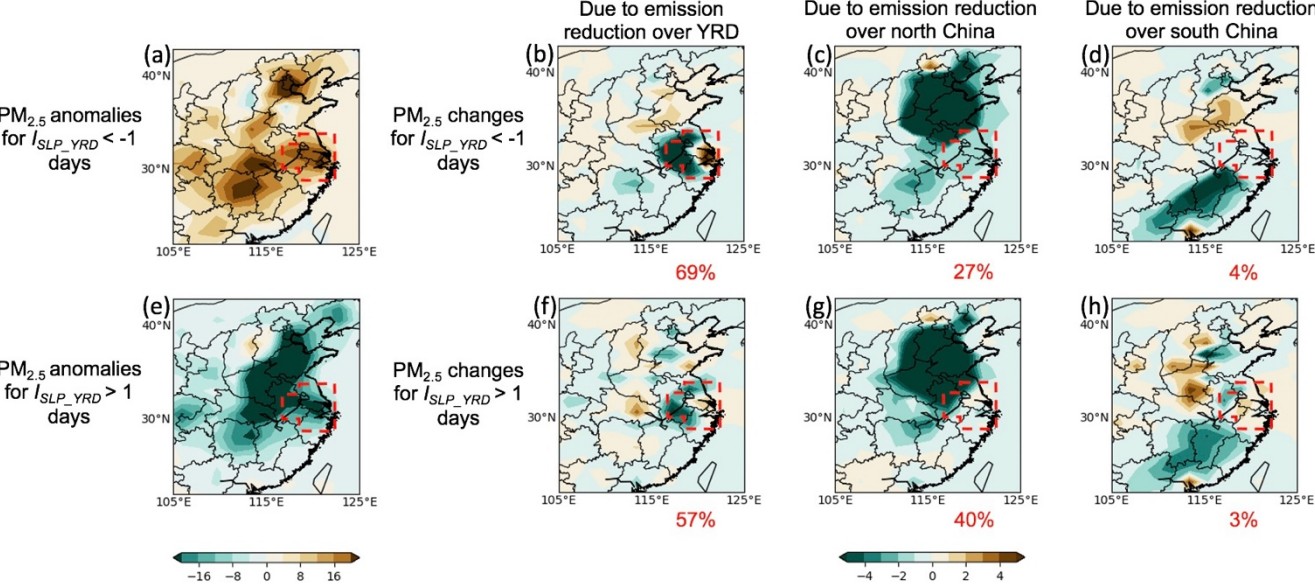

**Figure 9: PM$_{2.5}$ anomalies with respect to the winter mean (µg/m³) for (a) days with $I_{SLP\_YRD}$ < -1 and (e) days with $I_{SLP\_YRD}$ > 1 during DJF 2014-2018, and PM$_{2.5}$ changes (µg/m³) for the same days due to emission reductions over (b, f) YRD, (c, g) north China and (d, h) south China. The red box represents the YRD region. The relative shares of the total PM$_{2.5}$ reduction over YRD from local, north and south are labelled at the bottom right of panels b-d, f-h.**





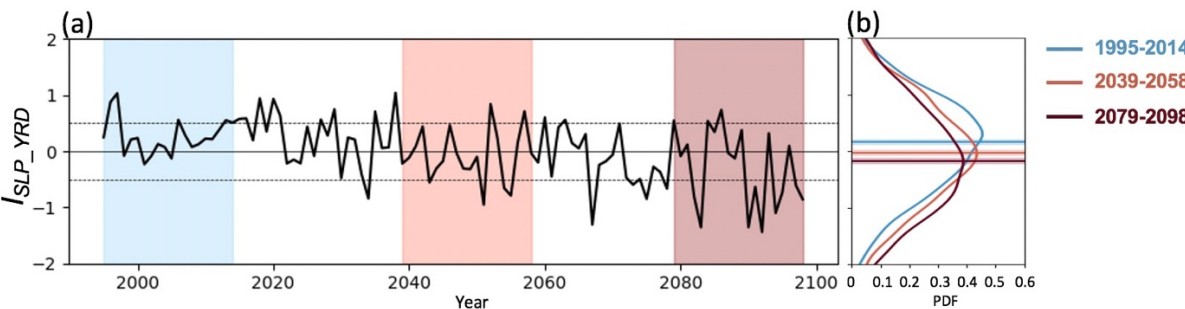

**Figure 10: (a) Time series of winter mean $I_{SLP\_YRD}$ from historical (1995–2014) and future (2015–2098, SSP3-7.0) simulations of UKESM1 in the CMIP6 archive. Blue, orange and red areas represent present day (1995-2014), mid-century (2039-2058) and the end of century (2079-2098), respectively. (b) Frequency distributions of daily mean $I_{SLP\_YRD}$ during winter over each period. The horizontal lines and shading represent the mean values and the associated 95% confidence intervals, respectively.**



**Table 1: Last six years of the nudged UKESM1 simulation for 1999-2019 (with CMIP6 historical emissions for year 2014) and three sensitivity simulations with reduced emissions (for year 2058 according the CMIP6 SSP3-7.0 scenario) over YRD, north China and south China, respectively. The three regions are displayed on Figure 8.**

| Year of meteorological data | Year of emission data | | |
|---|---|---|---|
| | YRD | north China | south China |
| 2014-2019 | 2014 | 2014 | 2014 |
| 2014-2019 | 2058 | 2014 | 2014 |
| 2014-2019 | 2014 | 2058 | 2014 |
| 2014-2019 | 2014 | 2014 | 2058 |

