# Peer review of "Modulation of daily PM2.5 concentrations over China in winter by large-scale circulation and climate change"

_Atmospheric Chemistry and Physics, 2022_

## Author Comment (AC1)

Dear ACP editor and reviewers,

We thank both reviewers for their positive comments and constructive suggestions. Below we provide our point-by-point replies to their comments (which are **in bold**).

**Referee: 1**

**(1) The results of this study are primarily based on UKESM1 model simulations. Although the authors compared the modeled PM2.5 concentrations with those from CAQRA reanalysis data, they have not verified whether the model could well reproduce the circulation conditions during the polluted days. The modeled PM2.5 only covers 20 years and the nationwide PM2.5 observations have already had 10-year data. Comparing the model simulation and observations in term of large-scale circulation and the index defined can give a more robust conclusion.**

In this study, we perform a nudged UKESM1 simulation, whereby the model is nudged with ERA-Interim reanalyses data for temperature and wind speed (see section 2.1). Therefore, we expect this nudged simulation to produce a realistic representation of the meteorological conditions. The figure below shows differences in meteorological variables between heavily polluted days over YRD identified with UKESM1 and the winter mean for ERA-Interim (top row) and UKESM1 (bottom row) during 2013–2017. The spatial patterns are very similar, indicating that UKESM1 can reproduce the circulation conditions during polluted days well. In order to establish robust relationships between the $PM_{2.5}$ concentrations and the atmospheric circulation, the UKESM1 model with emissions fixed at 2014 levels is used in this study. The downward trend (surely not linear following specific emission control measures) in the 10-year $PM_{2.5}$ observations would obscure such relationships and therefore complicate the analyses.

[Figure]

**(2) The future predictions of climate change impact of PM2.5 is investigated in this study based on changes in meteorological fields under SSP3-7.0 pathway. First of all, SSP3-7.0 is not a representative scenario for future air quality or climate change, at least in China. The scenario assumes the anthropogenic emissions of air pollutants continue to increase for a long time after 2015, but the emissions in China have significantly reduced since 2010s, which largely affect regional climate and cause the inaccurate of regional climate under SSP3-7.0. Also, China has committed to achieve carbon neutrality in 2060 and the results under the low forcing scenarios should be considered or discussed.**

Future changes in $PM_{2.5}$ precursor emissions are not considered in this study, as the focus is on the effects of circulation changes in a future climate. These are driven by changes in greenhouse gas concentrations and other climate forcers. SSP3-7.0 has been selected as it has a strong climate change signal which is likely to influence $PM_{2.5}$. Therefore, future changes in $PM_{2.5}$ concentrations over YRD projected in this study are driven purely by circulation changes under a high climate forcing scenario. We have also projected changes in the circulation-based index for YRD under two low forcing scenarios (Figs S10). Again these projections only consider the effect of climate change. The newly added Figure S10 shows the time series of winter mean $I_{SLP\_YRD}$ from historical (1995–2014) and future (2015–2098, SSP1-2.6 and SSP2-4.5) simulations of UKESM1 in the CMIP6 archive. The decrease in the strength of $I_{SLP\_YRD}$ and the increase in the interannual variability of $I_{SLP\_YRD}$ can also be projected under the low forcing scenarios, although the decrease in the strength of $I_{SLP\_YRD}$ is less dramatic than that under SSP3-7.0 (Fig. 10a). The mean value of $I_{SLP\_YRD}$ decreases from 0.14 (1995-2014) to -0.02 (2079–2098, SSP1-2.6), to -0.05 (2079–2098, SSP2-4.5), to -0.16 (2079–2098, SSP3-7.0). We have added this to the main text section 4 (revised manuscript page 4, lines 26-29), discussion and conclusions section (revised manuscript page 11, lines 29-34 and page 12, lines 1-2):

*"Future changes in $PM_{2.5}$ precursor emissions are not considered in this study, as the focus is on the effects of circulation changes in a future climate. These are driven by changes in greenhouse gas concentrations and other climate forcers. SSP3-7.0 has been selected as it has a strong climate change signal which is likely to influence $PM_{2.5}$."*

*"Future changes in $PM_{2.5}$ concentrations over YRD projected in this study are driven by circulation changes under a high climate forcing scenario (SSP3-7.0) and do not consider the effect of emissions of the main $PM_{2.5}$ components and precursors. We have also projected changes in the circulation-based index for YRD under two low climate forcing scenarios. Figure S10 shows the time series of winter mean $I_{SLP\_YRD}$ from historical (1995–2014) and future (2015–2098, SSP1-2.6 and SSP2-4.5) simulations of UKESM1 in the CMIP6 archive. An overall decrease in $I_{SLP\_YRD}$ and an increase in the interannual variability of $I_{SLP\_YRD}$ can also be projected under the low forcing scenarios, although the $I_{SLP\_YRD}$ decreases are less dramatic than that under SSP3-7.0 (Fig. 10a). The mean value of $I_{SLP\_YRD}$ is reduced from 0.14 in 1995–2014 to –0.02 (SSP1-2.6), –0.05 (SSP2-4.5) and –0.16 (SSP3-7.0) in 2079–2098."*

[Figure]

Figure S10: Time series of winter mean $I_{SLP\_YRD}$ from historical (1995–2014) and (a) future (2015–2098, SSP1-2.6) simulations, (c) future (2015–2098, SSP2-4.5) simulations of UKESM1 in the CMIP6 archive. Blue, orange and red areas represent present day (1995-2014), mid-century (2039-2058) and the end of century (2079-2098), respectively. (b) (d) Frequency distributions of daily mean $I_{SLP\_YRD}$ during winter over each period. The horizontal lines and shading represent the mean values and the associated 95% confidence intervals, respectively.

**(3) In addition, for the future predictions, how is the UKESM1 performed compared to other climate models in CMIP6. Different models tend to predict different regional circulation response. Does the conclusion that "a weaker pressure gradient between the Siberian High and the Maritime Continent Low" also exist in other CMIP6 models?**

Most CMIP5 and CMIP6 models show that the Siberian high weakens with additional warming (Miao et al., 2020; Zhao et al., 2021), although there are uncertainties in the projection of some features of the EAWM in CMIP5 models under small degrees of warming (e.g., 1.5 ℃). Therefore, we expect that a decrease in $I_{SLP\_YRD}$ representing a weaker pressure gradient between the Siberian High and the Maritime Continent Low is very likely to be simulated by other climate models as well showing at least 2 ℃ global warming, especially under high forcing scenarios. We now state this in the main text (revised manuscript page 12, lines 2-5):

*"A weakening of the Siberian high is simulated by most CMIP5 and CMIP6 models for global temperature increases of 2 ℃ or more (Miao et al., 2020; Zhao et al., 2021). Therefore, we expect that a decrease in $I_{SLP\_YRD}$ representing a weaker pressure gradient between the Siberian High and the Maritime Continent Low is very likely to be simulated by other climate models as well, especially under high forcing scenarios."*

**(4) Many studies have examined the circulation pattern and regional transport of air pollution over eastern China from the past to the future and they have similar or different conclusions. For example, Ren et al. (2021) quantified the sources of PM2.5 in many subregions of China and they found that PM2.5 pollution in eastern China is dominated by local emissions using an aerosol source tagging technique in an aerosol-climate model. Yang et al. (2021) examined the atmospheric circulation patterns conducive to severe haze in eastern China based on observations, modeling results and CMIP6 future predictions. They found that during the extreme pollution month the PM2.5 was mainly from aerosol transport from the North China Plain, although they also reported a future increase in the atmospheric circulation pattern conducive to the pollution under high forcing scenarios. Li et al. (2022) also highlighted the importance of climate change in regulating future air quality. They found that climate-driven aerosol changes are comparable to those contributed by changes in emissions over many regions of the world in high forcing scenarios. The authors are suggested to compare their results with previous studies.**

We have compared our results with previous studies and have now added some additional sentences in the main text to confirm that the results agree with these (revised manuscript page 8, lines 24-26 and page 11, lines 25-26):

*"This indicates that $PM_{2.5}$ pollution over YRD mainly originates locally (69%) and, to a lesser extent, from north China (31%), as found in other studies (e.g., Li et al., 2012; Ren et al., 2021)."*

*"This highlights the importance of climate-driven circulation changes in regulating future air quality, as found in previous studies (e.g., Pei et al., 2020, Yang et al., 2021)."*

**(5) Why only SSP3-7.0 scenario is selected?**

SSP3-7.0 is selected as it has a strong climate change signal, see our response to comment 2 above. Nevertheless, now we have also examined the future evolution of the circulation index for YRD under scenarios SSP1-2.6 and SSP2-4.5 (see changes in section 7 – provided above and Figure S10 of the revised manuscript- shown above).

**(6) Will the lack of SOA affect the conclusion, since that the circulation pattern is accompanied by temperature/relative humidity changes, affecting the formation of aerosols?**

UKESM1 does not include anthropogenic sources of SOA (see section 3), but it has a full treatment of SOA from biogenic sources (BVOCs). The yield of SOA generated from BVOCs is increased from 13% to 26% in the model to account for missing anthropogenic sources (Mulcahy et al., 2020). Therefore, SOA is not substantially underestimated, although it may not respond fully to changes in temperature or relative humidity as not all sources and formation mechanisms are accounted for. However, we expect the impacts of these uncertainties will not influence our main conclusions.

**(7) I noticed the authors have published a very similar paper in ACP using observations (Jia et al., 2022). They should clarify the new scientific findings in this study rather than the data used (model results and observations). The index defined in this study is not the same as Jia et al. (2022). Does that mean the model and observations will draw different results?**

In this study, an improved circulation index is used to characterize the relationship of the atmospheric circulation with the $PM_{2.5}$ concentrations (section 4), explain $PM_{2.5}$ sensitivity to emissions under different circulation conditions (section 5), and project future climate-driven changes in $PM_{2.5}$ concentrations (section 6). The difference in the circulation index defined between this study and Jia et al. (2022) partly comes from the longer time series available from the UKESM1 simulation (DJF 1999-2018) compared to that from CAQRA (DJF 2013-2017). In addition, the use of fixed emissions in UKESM1 allows establishing robust relationships between the atmospheric circulation and air pollution. The newly added Figure S2 (see below) shows that the heavily polluted days in YRD simulated by UKESM1 are mainly characterised by reduced SLP over eastern China for winter 2013-17, as found in Jia et al. (2022) for the same period, but are also characterised by enhanced SLP over the Maritime continent for the longer time period, 1999-2018. This suggests that the new SLP-gradient index, which takes a dipole structure over the Asian continent and the Maritime continent, encompasses the spatial variability in the large-scale circulation more completely. This index improves on the capability of the pressure index derived from the CAQRA reanalyses to distinguish $PM_{2.5}$ pollution levels in that region. We have now added further text on this in section 4 (revised manuscript page 7, lines 22-27):

*"We have checked that heavily polluted days in YRD simulated by UKESM1 are characterised by both reduced SLP over eastern China for winter 2013-17, as found in Jia et al. (2022) for the same period, and enhanced SLP over the Maritime continent for the longer*

*time period analysed here, i.e. 1999-2018 (Fig. S2). This indicates that the new SLP-gradient index, which takes a dipole structure over the Asian continent and the Maritime continent, encompasses the spatial variability in the large-scale circulation and its relationship with the winter PM$_{2.5}$ concentrations in YRD more completely.”*

[Figure]

Figure S2: Anomalies (heavily polluted days minus winter mean) of SLP (hPa, shading) and 850 hPa wind (m s$^{-1}$, vector) during (a) DJF 2013–2017 and (b) DJF 1999-2018 over YRD. Dotted regions mark statistically significant differences at the 95 % level. Grey shading represents the YRD region.

**(8) There are many uncertainties in this studies that should be discussed. I strongly recommend the authors to add a discussion section. For example, the model has biases in reproducing mean aerosol concentrations and the correlation coefficients reported in this study are not high enough. These may influence the results.**

We have renamed the "Conclusions" section to the "Discussion and conclusions" section with two more paragraphs. The first new paragraph briefly covers the replies to comments 2 and 3 above. The second new paragraph discusses the uncertainty in the UKESM1 simulated PM$_{2.5}$ concentrations (revised manuscript page 12, lines 7-16):

*"This study benefits from the state-of-the-art Earth system model UKESM1, but there is still some uncertainty in the simulated PM$_{2.5}$ concentrations. Like other global models, UKESM1 has a coarse horizontal resolution (1.875° in longitude and 1.25° in latitude) which limits the representation of regional meteorological fields (Chen et al. 2012; Zha et al., 2020; Xu et al., 2021) that depend on subgrid scale processes (e.g., relative humidity; surface wind speed). These may impact their ability to simulate secondary aerosol formation and growth and the ventilation of air pollutants. Moreover, PM$_{2.5}$ concentrations are generally underestimated in CMIP6 models (Turnock et al., 2020), including UKESM1, and this may be due to the absence or underrepresentation of some aerosol formation processes (e.g., nitrate and anthropogenic secondary organic aerosols). Nevertheless, the influence of the winter large-scale circulation on daily concentrations of PM$_{2.5}$ and the sensitivity to emissions found in this study should not be heavily impacted by this, as these results are based on the day-to-day variability of PM$_{2.5}$ concentrations rather than absolute PM$_{2.5}$ concentrations.”*

Regarding the correlation coefficients reported in this study, $I_{SLP\_YRD}$ is significantly correlated ($p < 0.01$) with PM$_{2.5}$ concentrations in YRD on daily time scales ($r = -0.47$) (see section 4). The non-linear relationship between $I_{SLP\_YRD}$ and PM$_{2.5}$ concentrations has also been acknowledged in section 6 before projecting future PM$_{2.5}$ concentrations (see comments about Fig. S8). There we wrote "As there is non-linearity and considerable spread, a resampling method is used …".

**(1) The authors published a highly relevant paper on ACP earlier this year, using the same model and a very similar analysis approach. However, the circulation-based indices defined in the two studies differ from one another, both in variable choices and key region detections. The authors claimed that the differences might be attributed to the different time spans. Does it mean that the index definition is sensitive to years and models? What do the uncertainties originate from, model instabilities or interannual-to-decadal variability of the relationship between PM2.5 and circulations? What are the dominant variables (e.g. SLP, V850, Z500, etc.) in characterizing the day-to-day variability of PM2.5 over different regions? Without a consistent definition of indices, the implication would be quite limited.**

The day-to-day variability of $PM_{2.5}$ can be characterized by different large-scale circulation variables because they are closely related. This is the reason why different authors have found different EAWM indices based on different circulation patterns. Therefore, the results of this study are not inconsistent with Jia et al. (2022). It is very likely that we have improved the representativeness of the indices in this study for simulation $PM_{2.5}$ concentrations because we have used a longer dataset alongside fixed emissions (see our response to comment 7 by referee 1 above).

**(2) The projections of the circulation-based indices in the future should be ensembled from more CMIP models to test the robustness of the trend. Besides, more SSP-RCP scenarios should be considered when evaluating the possible response of PM2.5 both to circulations and emissions, since the authors consider exploring the daily pollution responses to climate change in the title.**

We have now projected changes in the circulation-based index for YRD under low climate forcing scenarios (SSP1-2.6 and SSP2-4.5, newly added Fig S10). Please see the reply to comment 2 by referee 1 above.

However, examining other CMIP models besides UKESM1 is beyond the scope of this manuscript. Note, however, that similar changes in the atmospheric circulation can be expected for other models, especially under high forcing scenarios. See the reply to comment 3 by referee 1.

**(3) Why is the correlation map shown in Figure 2 different from Figure 1 in Jia et al., 2022, especially for YRD? The PM2.5 levels in YRD were closely connected with maritime airmasses in Jia et al., 2022, but correlated better with a broader region of inland in 1999-2018.**

The correlation map shown in Figure 2 is based on model results from UKESM1 for a 20-year winter period, while Figure 1 in the earlier paper is based on the CAQRA reanalysis for a shorter period of 5 winters. It is therefore not that surprising that they differ slightly. In both studies, only the highly correlated ($r \geq 0.7$) regions are used to represent YRD. The size of YRD identified in this study is slightly larger than that in the earlier study, with some westward expansion.

**(4) I notice that the model failed to capture some heavily polluted episodes for the three regions, even for the year 2014. Since the indices and composite analysis are all based on the model simulations, would it significantly impact the results?**

This is an interesting point. The underestimation of some heavily polluted episodes in UKESM1 could be due to a number of factors, including the absence or underrepresentation of some aerosol formation processes (e.g. nitrate), the coarse horizontal (and vertical) resolution of the atmospheric model and also the emission inventory used which is at the same resolution as the atmospheric model. The way the model treats emission profiling might also contribute to this underestimation. Nevertheless, the daily $PM_{2.5}$ concentrations from UKESM1 are shown to be significantly correlated (p < 0.01) with those from CAQRA, especially over YRD ($r = 0.60$) (Fig. 3). We therefore expect the results of this study should not be heavily impacted by the underestimation of some heavily polluted episodes. We have added a new paragraph that discusses the uncertainty in the UKESM1 simulated $PM_{2.5}$ concentrations (revised manuscript page 12, lines 7-16):

*"This study benefits from the state-of-the-art Earth system model UKESM1, but there is still some uncertainty in the simulated $PM_{2.5}$ concentrations. Like other global models, UKESM1 has a coarse horizontal resolution (1.875° in longitude and 1.25° in latitude) which limits the representation of regional meteorological fields (Chen et al. 2012; Zha et al., 2020; Xu et al., 2021) that depend on subgrid scale processes (e.g., relative humidity; surface wind speed). These may impact their ability to simulate secondary aerosol formation and growth and the ventilation of air pollutants. Moreover, $PM_{2.5}$ concentrations are generally underestimated in CMIP6 models (Turnock et al., 2020), including UKESM1, and this may be due to the absence or underrepresentation of some aerosol formation processes (e.g., nitrate and anthropogenic secondary organic aerosols). Nevertheless, the influence of the winter large-scale circulation on daily concentrations of $PM_{2.5}$ and the sensitivity to emissions found in this study should not be heavily impacted by this, as these results are based on the day-to-day variability of $PM_{2.5}$ concentrations rather than absolute $PM_{2.5}$ concentrations."*

**(5) In Line 20-23, Page 8, why are heavily polluted days impacted more strongly by emission reductions? In addition to the absolute levels change (from 3.1ug/m3 to 8.5 ug/m3 from local contribution for example), the relative values with respect to daily mean also increase (from 6.7% to 15.8% in this case). How to explain it?**

We thank the reviewer for this comment and I am happy to expand on our results. The daily mean $PM_{2.5}$ concentration on heavily polluted days is 53.9 μg/m³ higher than the winter mean value (46.9 μg/m³), so it is reasonable that the absolute levels change due to emission reductions is greater on heavily polluted days. Regarding the relative values with respect to the daily mean, local emissions contribute 6.6% (3.1/46.9) of the winter mean $PM_{2.5}$ concentration and 8.4% (8.5/(46.9+53.9)) of the mean $PM_{2.5}$ concentration on heavily polluted days. This suggests that local emissions slightly contribute more to $PM_{2.5}$ pollution over YRD on heavily polluted days, and this is consistent with slower wind speeds and greater stagnation on these days.

**(6) In section 4 the authors claimed that the index over YRD could "distinguish effectively between different levels of air pollution", especially for heavily polluted and clean conditions. While in section 5, the PM2.5 response to emission reductions in I<-1 days (Line 28, Page 8) shows similar values to the situation when considering winter**

**daily mean PM2.5 (Line 18, Page 8), but largely below the actual polluted conditions (Line 23, Page 8). How to address the contradiction?**

There is not a contradiction because as outlined by the reviewer in section 4, the index over YRD ($I_{SLP\_YRD}$) is shown to distinguish effectively between different levels of air pollution during DJF 1999-2018. However, the analyses in section 5 focus on regional contributions and are based on shorter, 6-year sensitivity simulations (2014-2019). During DJF 2014-2018, clean conditions mainly occur on days with $I_{SLP\_YRD} > 1$, which can be well distinguished from heavily polluted conditions that mainly occur on days with $-1 < I_{SLP\_YRD} < 0$, but the differences are not significant between heavily and moderately polluted days (Figure below).

[Figure]

In addition, the table below shows the relative shares of the total PM$_{2.5}$ decreases over YRD when emissions are reduced within the region and in the north. On days with $I_{SLP\_YRD} > 1$, PM$_{2.5}$ pollution over YRD originates more from the region (58%) than from north China (42%). As $I_{SLP\_YRD}$ decreases, the local contribution increases, and it reaches 73% and 72% on days with $-1 < I_{SLP\_YRD} < 0$ and $I_{SLP\_YRD} < -1$, respectively. The winter mean condition is more similar to those for days with $I_{SLP\_YRD} < -1$ than $I_{SLP\_YRD} > 1$. The relative shares of the total reduction from local and north on heavily polluted days are between those on days with $I_{SLP\_YRD} < 0$ and $0 < I_{SLP\_YRD} < 1$, which is consistent with the distributions of $I_{SLP\_YRD}$ for different percentile thresholds shown in the figure above.

| Relative shares of the total PM$_{2.5}$ reduction (PM$_{2.5}$ changes over YRD) | Due to emission reduction over YRD | Due to emission reduction over north China |
|---|---|---|
| $I_{SLP\_YRD} > 1$ days | 58% (-2.1 µg/m³) | 42% (-1.5 µg/m³) |
| $0 < I_{SLP\_YRD} < 1$ days | 66% (-2.5 µg/m³) | 34% (-1.3 µg/m³) |
| $-1 < I_{SLP\_YRD} < 0$ days | 73% (-3.8 µg/m³) | 27% (-1.4 µg/m³) |
| $I_{SLP\_YRD} < -1$ days | 72% (-3.6 µg/m³) | 28% (-1.4 µg/m³) |
| All winter days | 69% (-3.1 µg/m³) | 31% (-1.4 µg/m³) |
| Heavily polluted days | 71% (-8.5 µg/m³) | 29% (-3.4 µg/m³) |

**(7) In Figure 9(b) and 9(f), why do reductions in emission over YRD lead to a PM2.5 increase over coastal regions? The emission and compositions in PM2.5 changes are suggested to be investigated since it shows large nonlinearity.**

This is a good point and we have checked how emission changes affect the composition of $PM_{2.5}$. The newly added Figure S5 (see below) shows the increase in winter mean daily $PM_{2.5}$ concentrations over a coastal region in the east of YRD due to emission reductions over YRD (panel c). Among the major $PM_{2.5}$ components (i.e. organic matter (OM), sulphate, black carbon and sea salt), a great increase can only be found in OM concentrations over the same region (panel b). Winter mean CMIP6 emission changes from historical 2014 to SSP3-7.0 2058 are then investigated, and positive values can only be found for organic carbon (OC) from fossil fuel combustion (panel a). These results suggest that although the total emission of the main anthropogenic sources of $PM_{2.5}$ is reduced from historical 2014 to SSP3-7.0 2058 (Table S1), the emission of OC from fossil fuel combustion increases, especially over the east of YRD. This then leads to increases in the OM and $PM_{2.5}$ concentrations there. However, the results of this study should not be heavily impacted by this as we focus on the broader YRD region (red box in Figure S5) and the winter mean total emission of the main anthropogenic sources of $PM_{2.5}$ decreases by 41% from 2014 to 2058 for all three regions (YRD, north China and south China) (Table S1). We have now added further text on this to section 5 (revised manuscript page 9, lines 13-19):

*"Note that, although the daily mean $PM_{2.5}$ concentration averaged in YRD decreases due to emission reductions within the region, some increases can be found over the coast (eastern edge of red box in panels b and f of Figure 9). This is related to the increase in organic carbon emissions from fossil fuel combustion from historical 2014 to SSP3-7.0 2058 (Fig. S5a) that leads to increases in both organic matter (Fig. S5b) and $PM_{2.5}$ (Fig. S5c) concentrations there. However, the results of this study should not be heavily impacted by this as we focus on the broader YRD region where the winter emission changes of the main anthropogenic sources of $PM_{2.5}$ are dominated by a reduction in sulphur dioxide, leading to a total emission decrease of 41% from 2014 to 2058 (Table S1)."*

[Figure]

Figure S5: (a) CMIP6 emission changes of organic carbon (OC) from fossil fuel combustion from historical 2014 to SSP3-7.0 2058. Winter mean (b) organic matter (OM) changes ($\mu g/m^3$) and (c) $PM_{2.5}$ changes ($\mu g/m^3$) during DJF 2014-2018 due to emission reductions over YRD. The red box represents the YRD region.

**(8) Would it be possible to include the $PM_{2.5}$ projection in future scenarios in Figure 10 or an additional figure?**

We have now included the distribution of the climate-driven $PM_{2.5}$ concentrations derived from the historical and SSP3-7.0 scenarios in newly added Figure S9 (see below) as suggested. This figure is mentioned in section 6 (revised manuscript page 10, lines 19-20):

*"The mean PM$_{2.5}$ concentration increases from present day (46.3 μg/m$^3$) to mid-century (48.9 μg/m$^3$) and to the end of the century (50.1 μg/m$^3$) (Fig. S9)."*

[Figure]

Figure S9: Normal frequency distributions of daily mean climate-driven PM$_{2.5}$ concentrations (μg/m$^3$) over YRD during winter over present day (1995-2014), mid-century (2039-2058) and the end of century (2079-2098). The vertical lines and shading represent the mean values and the associated 95% confidence intervals, respectively.

References:

Chen, L., Pryor, S. C., & Li, D.: Assessing the performance of Intergovernmental Panel on Climate Change AR5 climate models in simulating and projecting wind speeds over China, Journal of Geophysical Research: Atmospheres, 117(D24), 2012.

Jia, Z., Doherty, R. M., Ordóñez, C., Li, C., Wild, O., Jain, S., and Tang, X.: The impact of large-scale circulation on daily fine particulate matter (PM2.5) over major populated regions of China in winter, Atmos. Chem. Phys., 22, 6471–6487, https://doi.org/10.5194/acp-22-6471-2022, 2022.

Li, M., Huang, X., Zhu, L., Li, J., Song, Y., Cai, X. and Xie, S.: Analysis of the transport pathways and potential sources of PM10 in Shanghai based on three methods. Science of the total environment, 414, 525-534, https://doi.org/10.1016/j.scitotenv.2011.10.054, 2012.

Miao, J., Wang, T. and Chen, D.: More robust changes in the East Asian winter monsoon from 1.5 to 2.0° C global warming targets, International Journal of Climatology, 40(11), 4731-4749, https://doi.org/10.1002/joc.6485, 2020.

Mulcahy, J. P., Johnson, C., Jones, C. G., Povey, A. C., Scott, C. E., Sellar, A., Turnock, S. T., Woodhouse, M. T., Abraham, N. L., Andrews, M. B., Bellouin, N., Browse, J., Carslaw, K. S., Dalvi, M., Folberth, G. A., Glover, M., Grosvenor, D. P., Hardacre, C., Hill, R., Johnson, B., Jones, A., Kipling, Z., Mann, G., Mollard, J., O'Connor, F. M., Palmiéri, J., Reddington, C., Rumbold, S. T., Richardson, M., Schutgens, N. A. J., Stier, P., Stringer, M., Tang, Y., Walton, J., Woodward, S., and Yool, A.: Description and evaluation of aerosol in UKESM1 and HadGEM3-GC3.1 CMIP6 historical simulations, Geosci. Model Dev., 13, 6383–6423, https://doi.org/10.5194/gmd-13-6383-2020, 2020.

Pei, L., Yan, Z., Chen, D., Miao, S.: Climate variability or anthropogenic emissions: which caused Beijing haze?, Environ. Res. Lett., 15, 3, p.034004, https://doi.org/10.1088/1748-9326/ab6f11, 2020.

Ren, L., Yang, Y., Wang, H., Wang, P., Chen, L., Zhu, J., and Liao, H.: Aerosol transport pathways and source attribution in China during the COVID-19 outbreak, Atmos. Chem. Phys., 21, 15431–15445, https://doi.org/10.5194/acp-21-15431-2021, 2021.

Turnock, S. T., Allen, R. J., Andrews, M., Bauer, S. E., Deushi, M., Emmons, L., Good, P., Horowitz, L., John, J. G., Michou, M., Nabat, P., Naik, V., Neubauer, D., O'Connor, F. M., Olivié, D., Oshima, N., Schulz, M., Sellar, A., Shim, S., Takemura, T., Tilmes, S., Tsigaridis, K., Wu, T., and Zhang, J.: Historical and future changes in air pollutants from CMIP6 models, Atmos. Chem. Phys., 20, 14547–14579, https://doi.org/10.5194/acp-20- 14547-2020, 2020.

Xu, Z., Han, Y., Tam, C. Y., Yang, Z. L., and Fu, C.: Bias-corrected CMIP6 global dataset for dynamical downscaling of the historical and future climate (1979–2100), Scientific Data, 8, 1–11, https://doi.org/10.1038/s41597-021-01079-3, 2021.

Yang, Y., Zhou, Y., Li, K., Wang, H., Ren, L., Zeng, L., Li, H., Wang, P., Li, B. and Liao, H.: Atmospheric circulation patterns conducive to severe haze in eastern China have shifted under climate change. Geophysical Research Letters, 48(23), p.e2021GL095011, 2021.

Zha, J., Wu, J., Zhao, D., and Fan, W.: Future projections of the near-surface wind speed over eastern China based on CMIP5 datasets, Clim. Dynam., 54, 2361–2385, https://doi.org/10.1029/2012JD017533, 2020.

Zhao, S., Feng, T., Tie, X., Li, G. and Cao, J.: Air pollution zone migrates south driven by East Asian winter monsoon and climate change, Geophysical Research Letters, 48(10), e2021GL092672, https://doi.org/10.1029/2021GL092672, 2021.

---

## Author Response (AR2)

Dear ACP editor and reviewers,

We thank both reviewers for their positive comments and constructive suggestions. Below we provide our point-by-point replies to their comments (which are **in bold**).

**Referee: 1**

**"Future changes in PM2.5 precursor emissions are not considered in this study, as the focus is on the effects of circulation changes in a future climate. These are driven by changes in greenhouse gas concentrations and other climate forcers." It should be causion to present as it is, since that other climate forcers also include aerosol precursor emissions. And in the polluted region like China, the role of aerosol variation on climate change could be comparable to, or even larger than, the greenhouse gas concentrations.**

We have rephrased this part in the main text (revised manuscript page 4, lines 26-29),

*"We use this strong climate change scenario to quantify how changes in climate alone are likely to affect $PM_{2.5}$ concentrations in the region. Note that the projected circulation changes will be affected by changes in the emissions of both greenhouse gases and aerosol precursors, while the direct contribution of precursor emission changes to the future evolution of the $PM_{2.5}$ concentrations is not considered in this study."*